# Ice Adhesion Evaluation of PTFE Solid Lubricant Film Applied on TiO$_2$ Coatings

**Emad Farahani** [1], **Andre C. Liberati** [1], **Amirhossein Mahdavi** [1], **Pantcho Stoyanov** [2], **Christian Moreau** [1] and **Ali Dolatabadi** [3,*]

1   Department of Mechanical, Industrial and Aerospace Engineering, Concordia University, Montreal, QC H3G 1M8, Canada
2   Department of Chemical and Materials Engineering, Concordia University, Montreal, QC H3G 1M8, Canada
3   Department of Mechanical and Industrial Engineering, University of Toronto, Toronto, ON M5S 3G8, Canada
*   Correspondence: ali.dolatabadi@utoronto.ca

**Abstract:** Ice formation affects the performance of many industrial components, including aircraft wings, spacecraft, and power transmission cables. In particular, ice build-up on airplane components increases drag and fuel consumption. A large number of studies have been carried out to reduce ice adhesion by developing passive methods such as icephobic coatings and active ice removal approaches such as mechanical vibrations or chemical-based solutions. Despite remarkable recent breakthroughs in the fabrication of icephobic coatings, passive ice removal solutions require higher durability to resist cyclical mechanical ice detachment treatments. Functionalized TiO$_2$ coatings, applied using the suspension plasma spray (SPS) technique, have been shown to be robust and to have dual-scale characteristics in an ice accretion analysis. In this study, the icephobicity and mechanical durability of a novel duplex coating consisting of polytetrafluoroethylene (PTFE) solid lubricant films on TiO$_2$-coated substrates were evaluated. Notably, various amounts of PTFE were applied on top of the TiO$_2$ coating to identify the ideal quantity required to obtain optimal icephobic properties. Ice was generated in an icing wind tunnel, and the amount of accreted ice was evaluated to assess the anti-icing properties. Wettability parameters, including static water contact angle and contact angle hysteresis, were measured to determine the water mobility and surface energy. Ice shear adhesion to the PTFE-TiO$_2$ duplex coating was measured using a custom-built test rig. The mechanical durability was assessed by measuring the ice shear strength for almost twenty icing–deicing cycles, and after five cycles, the roughness parameters and images taken from the surface of the samples were compared. The combination of PTFE solid lubricant film and TiO$_2$ coating reduced ice adhesion by 70%–90% compared to that of a bare aluminum substrate (reference material). Additionally, the results showed that the application of a uniform layer of PTFE solid lubricant film on dual-scale TiO$_2$ coating significantly reduced ice adhesion and maintained mechanical durability for 25 deicing cycles, making this combination a promising candidate for deicing approaches.

**Keywords:** ice adhesion; icephobicity; mechanical durability; TiO$_2$; PTFE solid lubricant

## 1. Introduction

Ice accretion is a significant issue in several industries, including power generation, transportation, renewable energy, communications, and aviation, since it can affect performance or place unwanted loads on structures [1]. In the aviation industry, it is a more challenging issue because ice adhesion can lead to various problems such as increased fuel consumption and lack of proper control of aerodynamic sections, which can lead to serious safety issues [2]. Passive and active methods are the two most popular approaches for dealing with ice problems [3]. Active deicing methods physically remove the ice using mechanical vibration solutions, scraping, and chemicals. However, passive anti-icing techniques such as icephobic coatings are more economically and environmentally reliable.

Active deicing methods are not preferred because they consume energy, add weight or complexity to the surface design, and affect structural integrity or performance [4]. Therefore, various passive coatings, including oil, slippery liquid-infused porous surfaces (SLIPSs), and superhydrophobic surfaces (SHSs) with nano patterns, have recently been applied [5]. Notably, SHSs help to limit the potential accumulation of water droplets on the surface of a substrate, which directly limits the potential for ice build-up at negative temperatures.

Different studies have shown that the hierarchical structure or dual-degree roughness of an SHS, where a nanostructure covers micro-sized features, can lead to the formation of the Cassie–Baxter wetting mode [6]. In this wetting mode, the SHS traps air bubbles in voids at the interface, preventing water droplets from penetrating the voids, and keeping them on top of its nanostructure. During freezing, the cavities prevent the droplets from forming a strong bond with the asperities, thereby acting as stress concentrators [7–9] that reduce ice adhesion [10]. Although most SHSs have exceptional water resistance and can maintain the Cassie–Baxter wetting mode, their mechanical durability after multiple icing–deicing cycles is controversial [11,12]. This is because icing–deicing cycles can damage the nanoscale properties of SHSs (e.g., reducing the nano-level roughness), and in addition, they are highly dependent on their micro- and nanostructure to maintain their superhydrophobic properties [12,13].

Several studies have employed different thermal spray methods to increase the durability of coatings [14,15]. As a versatile approach to producing hierarchical structures, suspension plasma spray has been used to deposit different oxides, such as $TiO_2$ and $SiO_2$ [16,17]. The hierarchical structure can increase water repellency on the surface and can provide a degree of mechanical durability for anti-icing purposes. These two properties are somewhat opposed to one another [15]. Sharifi et al. [18] produced an SHS by plasma spraying an ethanol-based suspension of $TiO_2$ nanoparticles, which resulted in improved water mobility and anti-icing durability of the $TiO_2$ coating [19]. More specifically, the SPS technique typically produces coatings with micron-sized cauliflower-like features, with nanometric particles that give the coatings two levels of roughness, dependent on the chosen scale of study; this is described as a dual-scale structure [18,19]. Furthermore, Brown et al. [20] deposited a 500 nm multilayer coating of DLC-$SiO_X$ on top of $TiO_2$ by chemical vapor deposition (CVD). Although they did not evaluate the mechanical durability by determining ice adhesion, they did measure surface wettability, and their coating showed improved durability after several icing–deicing cycles.

It has also been shown that SHS cannot act as an icephobic surface under all icing conditions. When the impact speed increases, the Bernoulli pressure of water droplets on the dual-scale surface roughness exceeds the critical capillary pressure between the asperities. The water droplets replace the air pockets, and the topographic features are eventually filled with water as they experience more pressure and higher impact velocity [21]. This can lead to strong mechanical interlocking at the interface, which results in high ice adhesion. To overcome this problem, some researchers have fabricated hydrophobic or superhydrophobic surfaces by applying a polymeric topcoat such as polytetrafluoroethylene (PTFE) or polydimethylsiloxane (PDMS) to a textured substrate, which improved the wettability and icephobicity when the surface was impacted by small water droplets at high speed [16,22–24]. Polymeric materials have shown high water contact angle (WCA) values, as an indication of decreased surface wettability, and low contact angle hysteresis (CAH) values, as an indication of water mobility on the surface [25]. More specifically, surfaces are considered to be hydrophobic when the WCA value is above 90°, while they are considered to be superhydrophobic when the WCA value is above 150°. Although it has been shown that polymeric coatings can increase the WCA value up to 120°, hierarchical roughness is necessary to improve stability and contact angles up to 150° [23]. Khaleghi et al. [17] used atmospheric plasma spray (APS) to deposit $TiO_2$ and $Al_2O_3$ on 316 L stainless steel. The WCA value of the combined coating was close to 56°, but was increased to 155° by adding a layer of PTFE. In addition, the application of a PTFE layer on a titania SHS [26] and PTFE-$SiO_2$ on glass [16], where a polymeric film covered a multiscale structure of the

plasma-sprayed coating, have shown promising water repellency results. These studies have mainly evaluated the wettability properties, and a reduction in ice adhesion was not reported. In a recent study by Farahani et al. [27], the mechanical durability of the formed ice on a PTFE solid lubricant film applied on an aluminum substrate was studied by employing a custom-built test rig. The results showed that ice adhesion was noticeably reduced when the PTFE solid lubricant film was used.

The above results confirm that the deposition of a polymeric hydrophobic film on a surface with a suitable dual-scale topology can be a practical approach for producing durable icephobic surfaces. However, uniform distribution of the hydrophobic polymer layer as a topcoat on a substrate with dual-scale roughness is needed to keep all the hierarchical features intact and to maintain the water-repellency property. Since PTFE solid lubricant has shown an acceptable reduction in ice adhesion, the aim of this study was to apply a uniform PTFE solid lubricant film on a $TiO_2$ coating to improve the anti-icing properties and durability. Accordingly, the WCA and CAH values were measured to assess the wettability characteristics. The weights of accreted ice on the PTFE-free $TiO_2$-coated (hereafter the "bare $TiO_2$ sample") and PTFE-coated samples were compared under different icing conditions in an ice wind tunnel (IWT). The icephobic characteristics of a thin layer of PTFE solid lubricant deposited on a $TiO_2$-coated substrate were evaluated. Before and after several cycles of ice detachment, surface roughness parameters and ice adhesion shear strength were evaluated to determine the mechanical durability of the coating.

## 2. Experimental Methods

### 2.1. Sample Preparation and PTFE Coating Application

To develop dual-scale surface roughness with a cauliflower-like structure, a $TiO_2$ coating was deposited on 13.5 mm × 40 mm × 3 mm 304 stainless steel substrates using the SPS technique with a 3MB plasma torch (Oerlikon Metco, Oerlikon Metco, Pfäffikon, Switzerland). The chosen feedstock material was 10 wt.% of submicron titanium dioxide powders (nominal average particle size of 500 nm, TKB Trading, Oakland, CA, USA) in an ethanol suspension. The complete suspension preparation process has been reported in detail elsewhere [18]. The stainless steel substrates were grit blasted prior to deposition in order to clean them, and therefore, improve coating adhesion. The chosen plasma gases were a combination of argon and hydrogen with an overall flow rate of 60 L/min. By setting the plasma current to 600 A and changing the hydrogen flow rate to make the plasma power 60 V, the power of the torch was adjusted to 36 kW. The suspension feed rate was 55 g/min, and the stand-off distance during the deposition was 5 cm [19]. The $TiO_2$-coated samples were washed in an ultrasonic bath after deposition, boiled in deionized distilled water to eliminate potential contamination, and dried in an air stream.

For reference substrates of Al, a thin film of PTFE solid lubricant (Everlube® R75, Curtiss-Wright, Davidson, NC, USA) was then applied as a topcoat by brushing and curing for 12 h, as described in [27]. This film is a solvent-based composite of PTFE with a polyamide-imide binder, made by polymerizing a large number of tetrafluoroethylene molecules, where fluorine atoms with high electronegativity make a strong bond with carbon in one difluoro methylene (CF2) group. In the presence of the binder, weak van der Waals bonds are formed between neighboring PTFE molecules. Since these weak forces in C-F are not polarizable, the result is a film with a high level of hydrophobicity and excellent lubricating properties [28,29].

Applying a defect-free and uniform PTFE solid lubricant film on the $TiO_2$-coated samples was challenging. It was necessary to apply a small amount of PTFE solid lubricant on top of the hierarchical structures of the $TiO_2$ coating to simultaneously maintain repellency and to fill the space between the asperities, to reduce ice adhesion when the water droplets penetrated the texture. With the nano-structured features of the $TiO_2$ coating, it is believed that the adhesion of the PTFE solid lubricant could be enhanced as compared to flat substrates [27], but this was not specifically studied here. A particular procedure, illustrated in Figure 1, was adopted to apply the PTFE solid lubricant film onto the $TiO_2$ coating; this procedure is similar to that employed by other researchers to deposit a thin coating layer onto a textured substrate [30,31]. First, a soft brush was used to apply a uniform quantity of PTFE solid lubricant film on top of the $TiO_2$ coating. Then, the samples were heated with a heating gun while undergoing vibrations in an ultrasonic bath. In this way, the distribution and coverage of the film over the rough $TiO_2$-coated surface was improved and the spaces between asperities could also be filled. Three types of samples were produced with this method, corresponding to different amounts of PTFE solid lubricant film being applied onto the $TiO_2$ coating. Samples with PTFE solid lubricant filling the asperities of the $TiO_2$ coating, but not covering the top of the asperities, were considered to be "partially filled" samples. Samples with PTFE forming a uniform layer that covered the tops and bottoms of the asperities were considered to be "filled" samples. Finally, samples with PTFE covering the top of the cauliflower-like features were considered to be "overfilled" samples. The different types of samples are illustrated in Table 1. Six partially filled and filled samples were also produced under different heating and vibration times, as shown in Table 1. The filled sample PTFE-Fil-6 was produced without heating. The overfilled samples were produced by applying the PTFE solid lubricants onto the $TiO_2$ coating without vibrations or heating. The sample PTFE-OVE-8 was produced by adding a second layer of PTFE after the air drying of the first layer. This was done to assess the possibility of a durable film if PTFE solid lubricant was applied in excess.

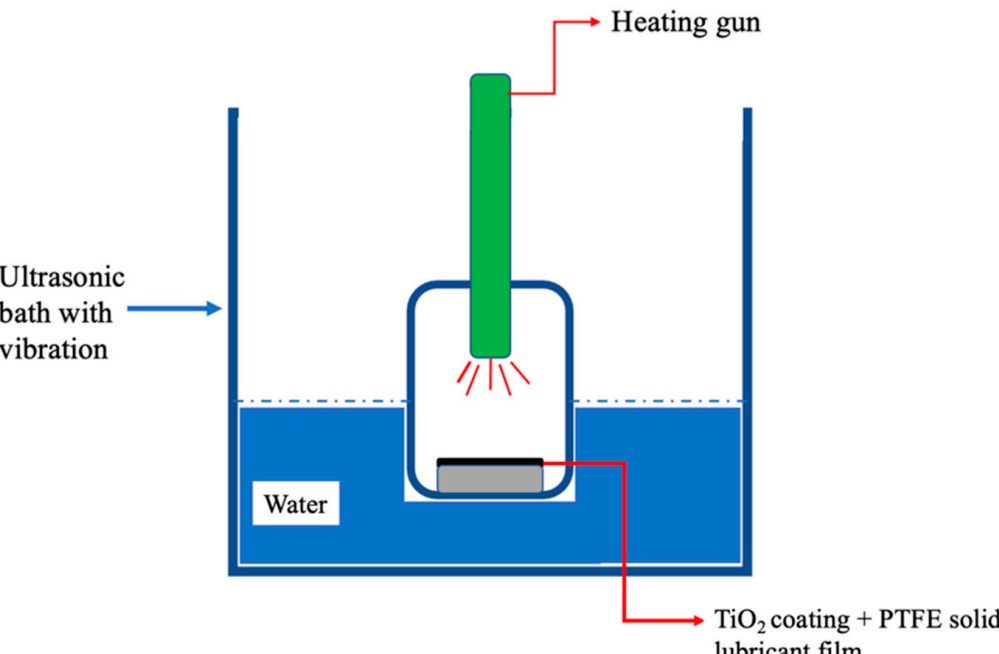

**Figure 1.** Schematic diagram of the application of PTFE solid lubricant film onto a $TiO_2$ coating by heating and vibration.

**Table 1.** Sample preparation details and their PTFE film distribution.

| Name | Film Distribution | Schematic | Vibration Time (Min.) | Heating Time (Min.) |
|---|---|---|---|---|
| PTFE-ParFil-1 | Partially Filled | | 20 | 20 |
| PTFE-ParFil-2 | Partially Filled | | 15 | 15 |
| PTFE-Fil-3 | Filled | | 10 | 10 |
| PTFE-Fil-4 | Filled | | 10 | 5 |
| PTFE-Fil-5 | Filled | | 5 | 5 |
| PTFE-Fil-6 | Filled | | 5 | 0 |
| PTFE-OVE-7 | Over Filled | | 0 | 0 |
| PTFE-OVE-8 | Over Filled + extra layer of PTFE film | | 0 | 0 |

## 2.2. Surface Characterization

In order to examine the surface properties, optical images of the bare $TiO_2$ and PTFE-coated samples were taken using a confocal microscope (LEXT OLS 4100, Olympus Corporation, Tokyo, Japan), since roughness parameters can directly affect ice adhesion [32]. Surface roughness parameters were also measured using the same confocal microscope with a 20× objective magnification on 2 mm × 2 mm block areas, according to ISO 4288 and ISO 25178-3. Furthermore, the average roughness was measured across a 50 μm × 50 μm area to verify the existence of dual-scale surface features, before and after the film deposition. Several roughness parameters, such as root mean square roughness ($R_q$), average roughness ($R_a$), skewness ($R_{sk}$), and kurtosis ($R_{ku}$), were measured, using the confocal microscope to analyze their effects on film distribution and ice adhesion properly. $R_a$ is determined by averaging the surface height distribution (peaks and valleys). The root mean square of the surface height distribution on a surface is denoted by $R_q$. Due to the nature of the root mean square, $R_q$ includes information on the average roughness ($R_a$) as well as the standard deviation of the height profile. The skewness ($R_{sk}$) of a surface is the asymmetry of the profile around the mean line. Positive skewness values imply that there are more valleys than peaks, while negative values mean there are more peaks. The distribution of peaks and valleys on the surface can be evaluated using the kurtosis ($R_{ku}$) parameter. When $R_{ku}$ is three, the distribution is normal (Gaussian). In addition, the average value of the absolute distance between the height of the five highest profile peaks (from $R_{p1}$ to $R_{p5}$) plus the five deepest valleys (from $R_{v1}$ to $R_{v5}$) is calculated as $R_z$, known as the peak-to-valley distance [33].

The WCA and CAH values were measured using the sessile droplet method at room temperature to determine surface wettability and water mobility, respectively [18]. It should be noted that the CAH is the difference between the advancing contact angle (ACA) and receding contact angle (RCA) of a water droplet on a surface. A dispenser was used to produce 10 μL water droplets of deionized water on a horizontal surface, released with zero velocity. The images of WCA and CAH were captured using a high-speed camera (Photron Ltd., Tokyo, Japan) under a backlight LED. Then, WCA and CAH values were calculated using the software ImageJ (NIH, Bethesda, MD, USA) [34]. The measurements were repeated five times on each sample to ensure the accuracy of the roughness data.

### 2.3. Anti-Icing Performance Analysis

An IWT was utilized for ice development as it simulates the impact of in-flight icing conditions. The details of the IWT performance and the measurement methods of its icing parameters, such as liquid water content (LWC) and median volume diameter of droplets (MVD), have been discussed elsewhere [35]. Three distinct types of testing were carried out on the various samples to assess their anti-icing performance, and they are presented in the following paragraphs. In addition, the surface temperature of the samples was measured using an infrared (IR) camera (FLIR A320, Teledyne FLIR LLC, Wilsonville, OR, USA) before and during ice development, within the IWT, to investigate how the PTFE solid lubricant film could delay temperature reduction of the substrate.

The first test was carried out to assess how well the PTFE solid lubricant film reduced the amount of accreted ice compared to the bare $TiO_2$ coating. In this test, the ice growth was limited to one minute, since longer durations led to ice layers accumulating on top of one another rather than on the substrate, and the properties of the substrate had no effect on the amount of accreted ice beyond that point. The test was conducted under eight distinct circumstances using airspeeds of $45 \pm 2$ m/s and $25 \pm 1$ m/s, temperatures of $-10$ °C and $-3$ °C, and angles of impact (AOI) of 45° and 90°. Under these conditions, clear and mixed types of ice can form [36]. As mentioned in [27], clear-type ice forms at temperatures close to 0 °C; droplets of supercooled water hit the surface but do not instantly freeze upon impact and spread across the surface while maintaining a liquid state. The liquid droplets diffuse into the asperities of the surface after impact and develop bonds with the valleys and peaks on the surface, resulting in strong mechanical interlocking. However, by gradually lowering the temperature, mixed ice ($-3$ to $-6$ °C), rime-mixed ice ($-7$ to $-10$ °C), and fully rime ice ($-20$ °C) can develop, and each of these has different strengths and structures compared to clear ice [37,38]. In the case of mixed ice, the droplets instantly solidify upon impact, entrapping air pockets that lead to several internal flaws and a relatively weaker structure [39,40]. The first tests were repeated nine times and the weight measurement of the accumulated ice on each sample was performed and compared each time.

The second test was carried out to evaluate the ice shear adhesion of the PTFE solid lubricant film in the most severe icing conditions identified in the first test, as shown in Table 2; these conditions correspond to the formation of clear ice, which is considered to be the most difficult type of ice to remove. The second tests were conducted for 180 s, at a temperature of $-3$ °C, an airspeed of 25 m/s, with an MVD of 30 μm and LWC of 0.8 g/m$^3$, which contributed to generating 2 to 3 mm of ice on the surface of the samples. At first, the ice developed on the sample within the IWT. Then, the sample was transferred to a custom-built test rig to detach the ice in less than 20 s. The shear test setup (Figure 1 of [27]) was conducted by using a motorized linear stage equipped with a force gauge (NEXTECH DFS-1000 series, Nextech Global Co. Ltd., Bangkok, Thailand) at a speed of 0.5 mm/s to detach the ice. More details on the test rig can be found in [27]. The force was supplied to the ice around 1 mm above the interface to achieve pure shear stress [41]. The maximum adhesive shear force (N) to detach the ice was divided by the contact area of the ice/substrate interface (mm$^2$) to calculate ice shear adhesion strength (kPa). The measurement was carried out nine times for each sample to ensure its repeatability. Finally, the adhesion reduction factor (ARF), being a well-known tool for comparing various adhesion methodologies across different studies [42], was also measured by dividing the ice adhesion on bare aluminum (as a reference) by the results of ice shear adhesion on the samples of this study.

**Table 2.** IWT parameters for shear adhesion and mechanical durability tests.

| Temperature (°C) | Airspeed (m/s) | MVD (μm) | LWC (g/m$^3$) | Time (s) | Ice Thickness (mm) |
|---|---|---|---|---|---|
| $-3$ | $25 \pm 3$ | $30 \pm 3$ | $0.8 \pm 0.02$ | 180 | 2 to 3 |

The aim of the third test was to assess the mechanical durability of the PTFE solid lubricant film under the icing parameters given in Table 2 (clear ice formation). Each sample was subjected to numerous ice production cycles within the IWT, and the ice was detached from the surface using the shear adhesion test equipment. After every five cycles, roughness was measured, and the physical state of the film was examined to detect damage and delamination. In addition, ice adhesion was measured after each icing cycle to specify the mechanical durability of the PTFE film.

## 3. Results

### 3.1. Film Distribution

The $TiO_2$ coatings that were deposited using the SPS technique produced a porous surface with cauliflower-like features, as illustrated by prior work performed by the research group of [19] (Figure 3a,b in reference [19]). Note that specific characterization of these coatings was not performed here, as the SPS coating development was not the focus of this study.

Figure 2 illustrates the surface images of three groups of samples after applying PTFE solid lubricants on the $TiO_2$ coating, under various vibration and heating circumstances. It appears that the film was distributed differently across the asperities and on top of the hierarchical features. Figure 2a shows the partially filled samples, and while the majority of the valleys were filled, several cauliflower tips were only partially covered, as seen in yellow. In the filled samples (Figure 2b), the spaces between surface features were fully covered, and the top of the cauliflower structure had a thin, uniform layer of PTFE solid lubricant. Finally, in the overfilled samples (Figure 2c), the asperities were fully covered by the PTFE solid lubricant film and produced a bumpy surface.

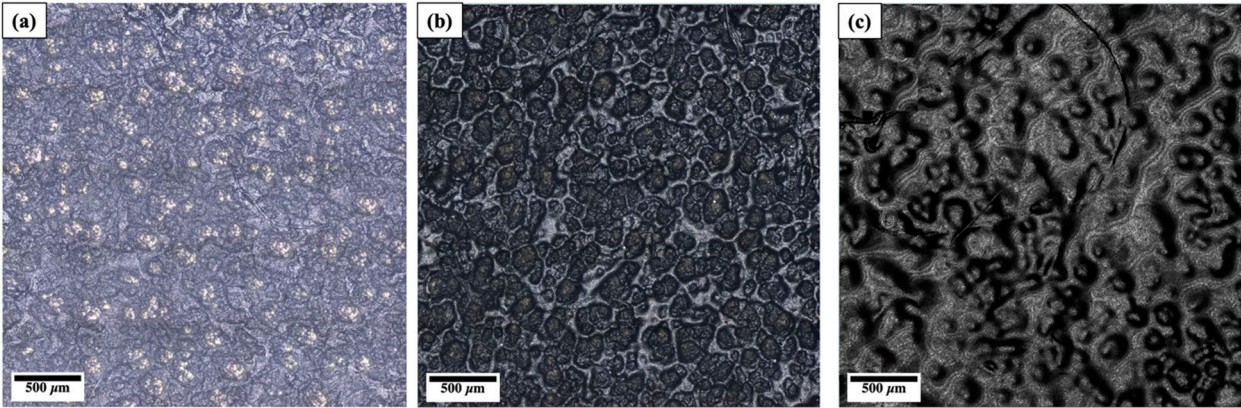

**Figure 2.** Surface pictures of: (**a**) Partially filled, (**b**) filled, and (**c**) overfilled samples.

### 3.2. Surface Wettability Characterization

The measured WCA (indicating static surface wettability) and CAH (indicating the water mobility) values for the bare $TiO_2$ surface and the PTFE-coated samples are given in Table 3. These values were obtained by repeating the measurements five times on each sample to ensure the accuracy of the roughness data. Since the $TiO_2$-coated surfaces are hydrophilic (WCA = 10°), water mobility is limited so it is challenging to measure the CAH; CAH can be measured only for hydrophobic and superhydrophobic surfaces with high WCA values.

Table 3 shows that the highest WCA value (131°) and the lowest CAH value (22°) are related to the filled samples. The WCA values of the partially filled and overfilled samples were 77° and 102°, respectively, and their CAH values were 103° and 63°, respectively. Despite the fact that neither the partially filled nor the overfilled samples indicated any discernible superhydrophobicity, they can still be classified as hydrophobic. Due to the fact that $TiO_2$ features are hydrophilic (Table 3), the wettability and hydrophobicity of the partially filled samples might be affected by the partially covered structure of the $TiO_2$

coating. However, it is worth noting that the measured values of WCA and CAH for the overfilled samples were between the other two samples, and that these values were comparable to the wettability of the PTFE solid lubricant film on a flat surface.

**Table 3.** WCA and CAH values of PTFE solid lubricant samples and TiO$_2$ coating.

| | Sample Name | Schematic | WCA (°) | CAH (°) |
|---|---|---|---|---|
| TiO$_2$ | ---------- | ---------- | $10 \pm 2$ | ---------- |
| TiO$_2$ + PTFE | Partially Filled | | $77 \pm 2$ | $103 \pm 1$ |
| TiO$_2$ + PTFE | Filled | | $131 \pm 7$ | $22 \pm 0.4$ |
| TiO$_2$ + PTFE | Over Filled | | $102 \pm 3$ | $63 \pm 1$ |

### 3.3. Surface Roughness Parameters

Table 4 shows the various surface roughness parameters of the samples. The highest R$_z$, R$_a$, and R$_q$ values are related to the bare TiO$_2$ sample, indicating how the spraying process can result in a rough surface. Moreover, the R$_z$, R$_a$, and R$_q$ values of the partially filled samples are lower than those of the bare TiO$_2$ sample. The difference in surface roughness between PTFE-Parfil-1 and PTFE-Parfil-2 indicates that heating and 15 min of vibration (Table 1) might have led to a change in the dispersion of the PTFE film. This can be put in parallel to Figure 2a that reveals that some cauliflowers are not fully covered by the PTFE film and are still visible (yellow dots).

**Table 4.** Roughness parameters of different samples.

| Name | Category | R$_p$ (µm) | R$_v$ (µm) | R$_z$ (µm) | R$_a$ (µm) | R$_q$ (µm) | R$_{Sk}$ (µm) | R$_{ku}$ (µm) |
|---|---|---|---|---|---|---|---|---|
| TiO$_2$ | ----------- | $15.5 \pm 0.2$ | $13.4 \pm 0.2$ | $28.9 \pm 0.9$ | $6.5 \pm 0.1$ | $8.1 \pm 0.1$ | $0.39 \pm 0.02$ | $3.2 \pm 0.03$ |
| PTFE-ParFil-1 | Partially Filled | $10.1 \pm 0.3$ | $8.9 \pm 0.3$ | $18 \pm 0.2$ | $4.2 \pm 0.2$ | $5.1 \pm 0.7$ | $0.16 \pm 0.01$ | $2.4 \pm 0.02$ |
| PTFE-ParFil-2 | Partially Filled | $9.5 \pm 0.2$ | $7.8 \pm 0.2$ | $17.3 \pm 0.2$ | $3.8 \pm 0.07$ | $4.6 \pm 0.09$ | $0.13 \pm 0.02$ | $2.5 \pm 0.02$ |
| PTFE-Fil-3 | Filled | $8.6 \pm 0.2$ | $5.2 \pm 0.1$ | $13.8 \pm 0.4$ | $2.1 \pm 0.04$ | $2.6 \pm 0.05$ | $0.05 \pm 0.01$ | $2.4 \pm 0.01$ |
| PTFE-Fil-4 | Filled | $6.9 \pm 0.2$ | $4.8 \pm 0.2$ | $11.7 \pm 0.4$ | $2 \pm 0.05$ | $2.4 \pm 0.02$ | $0.04 \pm 0.01$ | $2.9 \pm 0.01$ |
| PTFE-Fil-5 | Filled | $5.9 \pm 0.1$ | $3.9 \pm 0.1$ | $9.8 \pm 0.2$ | $1.4 \pm 0.04$ | $2 \pm 0.08$ | $0.03 \pm 0.01$ | $2.7 \pm 0.02$ |
| PTFE-Fil-6 | Filled | $3.2 \pm 0.2$ | $1 \pm 0.1$ | $4.2 \pm 0.4$ | $1.1 \pm 0.01$ | $1.4 \pm 0.02$ | $0.05 \pm 0.01$ | $2.9 \pm 0.01$ |
| PTFE OVE-7 | Over Filled | $1 \pm 0.1$ | $1.8 \pm 0.1$ | $2.8 \pm 0.5$ | $1 \pm 0.08$ | $1.2 \pm 0.08$ | $-0.05 \pm 0.01$ | $2.8 \pm 0.01$ |
| PTFE-OVE-8 | Over Filled | $0.3 \pm 0.1$ | $0.6 \pm 0.1$ | $0.9 \pm 0.1$ | $0.6 \pm 0.02$ | $0.8 \pm 0.03$ | $-0.02 \pm 0.01$ | $2.9 \pm 0.03$ |

The difference between R$_v$ and R$_p$ of the filled samples (from PTFE-Fil-3 to PTFE-Fil-6) is lower than that of the partially filled samples, leading to smaller R$_Z$ values and indicating a better distribution of the PTFE solid lubricant film. This might be due to the fact that shorter vibration and heating time (Table 1) can facilitate the movement of film from the peaks of the asperities into the valleys between the cauliflowers, while a thin homogeneous layer of the PTFE solid lubricant film is available on their surface, as shown in Figure 2b. The better film distribution of the filled samples may lead to smaller R$_a$ and R$_q$ values than partially filled and bare TiO$_2$ samples (Table 4).

In addition, the roughness parameters of the overfilled samples showed a significant decrease in R$_z$, R$_a$, and R$_q$ values and this can be associated with the full coverage of cauliflower features, which produced an uneven and bumpy surface. Figure 2c shows the formation of a thick layer of the solid lubricant film that completely covered almost all asperities.

Two additional roughness characteristics that were studied as effective parameters for ice adhesion are $R_{sk}$ and $R_{ku}$ [43]. $R_{sk}$ values larger than zero indicate that the surface might have more peaks than valleys. The PTFE-Fil-6 sample showed the lowest positive $R_{sk}$ value, and its $R_{ku}$ value was close to three, while the $R_{ku}$ values of the overfilled samples were very close to three, and their $R_{sk}$ values were negative.

Comparing the dual-scale roughness properties of the $TiO_2$ coating before and after PTFE deposition was another benefit of the evaluation of roughness parameters. As shown in Figure 3, the average roughness ($R_a$) on top of the cauliflower tips (regions b) and the spaces between them (region a) were measured and averaged for several different points in the sample before and after deposition. The $R_a$ values on the top and between the cauliflowers of the bare $TiO_2$ sample were $8 \pm 2$ μm and $4 \pm 1$ μm, respectively. They were dropped to $2 \pm 0.5$ μm and $0.8 \pm 0.1$ μm, respectively, after the PTFE deposition, indicating that the dual roughness and the nanostructure cauliflowers were both preserved.

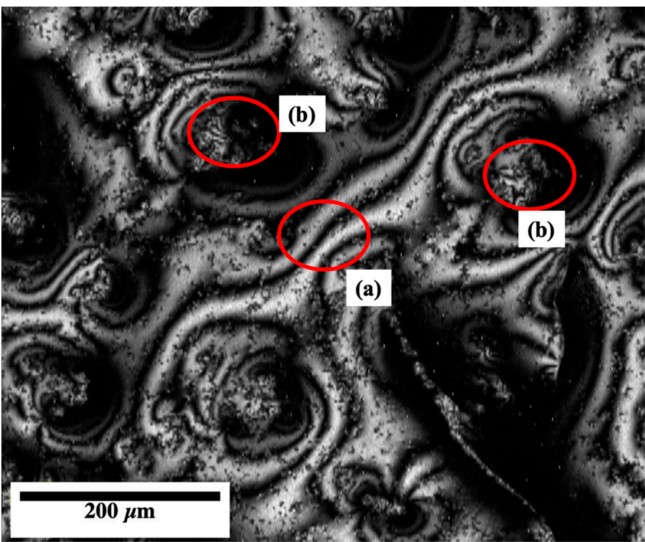

**Figure 3.** Dual-scale roughness evaluation: (**a**) Between cauliflowers; (**b**) on top of cauliflowers.

### 3.4. Delay in Substrate Temperature Reduction

Given its non-conductive nature, the effect of the PTFE solid lubricant film on the delay in surface temperature reduction was investigated. Figure 4 shows the plot of substrate temperature reduction (°C) against time (s) for the bare $TiO_2$ and the filled PTFE-coated samples within the IWT, as measured with an infrared camera.

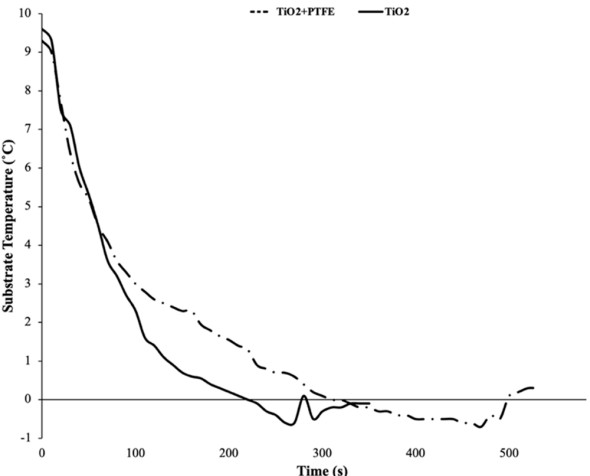

**Figure 4.** Substrate temperature reduction rate for the bare $TiO_2$-coated and the filled PTFE-coated samples.



The filled PTFE solid lubricant film led to a 60% reduction in the temperature rate and longer freezing delay time, providing more mobility for water droplets and a lower contact area of droplets on the substrate during the freezing procedure. The average temperature reduction rates for the bare $TiO_2$ and the PTFE-coated samples were 0.034 °C/s and 0.013 °C/s, respectively. This difference can be noticeably observed in Figure 4, where a delay of 100 s between the bare $TiO_2$ sample and the PTFE solid lubricant film is observed, to obtain a surface temperature of −1 °C. The effect of PTFE on the surface temperature delay (results of Figure 4) was consistently repeatable in this study, as well as in good agreement with other studies [44], which was due to the low thermal diffusivity and heat transfer coefficient of the PTFE [24] and the dual-scale roughness of the $TiO_2$ coating [45,46].

*3.5. Ice Accretion*

The total weight of the accreted ice was measured under eight different icing conditions to compare the anti-icing performance of the filled samples to that of the bare $TiO_2$ sample. The filled samples were selected since they seemed to be a better candidate for reducing the ice adhesion based on their higher WCA and lower CAH (Table 3). For each sample, the experiment was repeated nine times under each set of icing conditions, and the average weight is reported in Table 5.

**Table 5.** Ice accretion test results.

| No. of Icing Condition | Airstream Velocity (m/s) | Airstream Temperature (°C) | AOI (°) | Ice Type | Weight of Ice on $TiO_2$ Sample (g) | Weight of Ice on Filled PTFE Sample (g) | Icing Reduction (%) |
|---|---|---|---|---|---|---|---|
| 1 | 25 | −3 | 45 | Clear | 0.80 ± 0.07 | 0.24 ± 0.01 | 69 |
| 2 | 25 | −3 | 90 | Clear | 1.10 ± 0.04 | 0.31 ± 0.01 | 70 |
| 3 | 25 | −10 | 45 | Rime-mixed | 0.26 ± 0.01 | 0.08 ± 0.01 | 70 |
| 4 | 25 | −10 | 90 | Rime-mixed | 0.47 ± 0.04 | 0.14 ± 0.01 | 71 |
| 5 | 45 | −3 | 45 | Clear | 0.85 ± 0.02 | 0.17 ± 0.01 | 80 |
| 6 | 45 | −3 | 90 | Clear | 1.57 ± 0.01 | 0.41 ± 0.01 | 74 |
| 7 | 45 | −10 | 45 | Rime-mixed | 0.29 ± 0.01 | 0.07 ± 0.01 | 76 |
| 8 | 45 | −10 | 90 | Rime-mixed | 0.75 ± 0.03 | 0.20 ± 0.01 | 73 |

The PTFE solid lubricant coating considerably reduced the amount of accreted ice, with the reduction varying from 70 to 80 percent depending on the icing conditions. The higher weight reductions were observed at the highest airspeed (45 m/s) under icing conditions 5–8. Icing condition No. 5, with clear ice, showed the highest reduction in the accumulated ice (80%) under the highest air velocity (45 m/s) and temperature (−3 °C), and the lowest AOI (45°). Other icing conditions also showed similar behavior with a 76% reduction, including icing condition No. 7, which was performed at the same airspeed and AIO as No. 5 but at −10 °C with mixed ice. Lower ice accumulation was predictable since the filled samples showed hydrophobicity and delay in surface temperature reduction. As a result, water droplets had a higher chance of being blown away and separated by the high-speed airstream and wind drag force before freezing. Although there was less ice reduction under icing conditions No. 1–4 (at a lower airspeed), a significant reduction with a minimum value of 69% was still observed.

It should be noted that at the same volume, rime-mixed or mixed ice is lighter than clear ice due to existing internal cavities and lower density. The effect of the PTFE solid lubricant film on ice accretion reduction is promising, especially at a temperature of −3 °C where clear ice is formed, considering the fact that clear ice is known to be the most hazardous and difficult ice to detach due to its high strength and low bending [37], and the icing conditions were comparable to what can occur on the aerodynamic surface of an aircraft inflight.

Moreover, the total amount of accreted ice increased when the impact angle changed from 45° to 90°, especially at higher airspeed. The AOI of 45° and high speed facilitate the rolling of water droplets and lead to lower ice accumulation. However, with an AOI

of 90°, i.e., the worst-case scenario, the PTFE solid lubricant film performed better in all circumstances and lowered the weight of ice by at least 70%.

### 3.6. Shear Adhesion Strength of Ice

The shear adhesion strength of the ice was also assessed to analyze the anti-icing performance of the PTFE solid lubricant film. Figure 5 shows the results of ice adhesive shear strength for several samples, and the error bars indicate the standard deviation for nine performed measurements. The results show that the ice adhesion of the PTFE solid lubricant film on $TiO_2$ coating was significantly lower than that on bare Al, for all PTFE preparation methods.

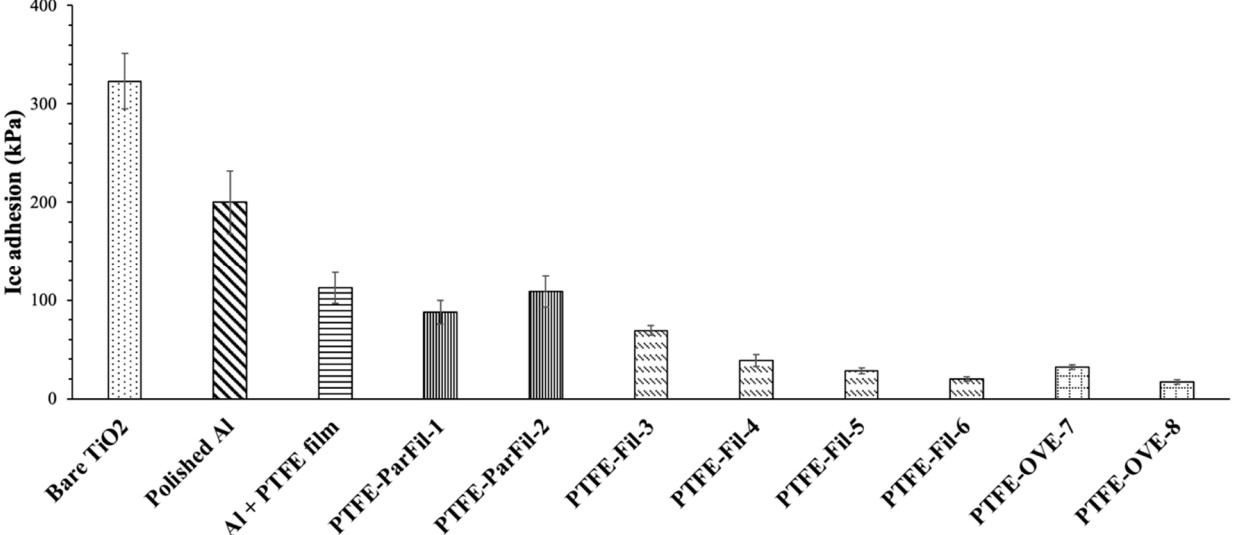

**Figure 5.** Ice shear adhesion strength on different samples, tested at −3 °C.

The bare $TiO_2$ sample had the highest ice shear adhesion with a large variation (323 ± 28 kPa). It was anticipated that the bare $TiO_2$ sample would exhibit high ice adhesion due to its high Ra and Rq values (Table 4) and its lower WCA (Table 3). In all icing scenarios, the accreted ice on the bare $TiO_2$ sample resulted in a substantial layer of accumulated ice (Table 5).

Additionally, on the one hand, the partially filled samples, with adhesion values of 87 ± 12 kPa and 108.6 ± 16 kPa and a corresponding ARF of about two, exhibited good icephobic properties, since icephobic surfaces are defined by adhesion values below 100 kPa [13]. Moreover, compared to the bare $TiO_2$ sample, they showed a considerable reduction of more than 60% in ice adhesion.

On the other hand, the overfilled samples showed considerably low ice adhesion, particularly PTFE-OVE-8 with 17.5 kPa. They had a significant ARF of roughly 10, highlighting the best PTFE solid lubricant performance as an icephobic coating for adhesion reduction. In previous work, the adhesion strength of PTFE solid lubricant films on bare Al substrates was found to be at 120 kPa (Figure 5), which leads to an ARF of around two (similar to what was obtained in [27], which shows that the overfilled PTFE film on $TiO_2$ structure can help to achieve a reduction in ice adhesion that is five times greater than PTFE solid lubricant placed directly on the substrate.

In addition, all filled samples (highest WCA, lowest CAH, and relatively low $R_a$) showed a low ice adhesion strength with noticeably low variations, which could be due to a homogeneous distribution of PTFE solid lubricant between the asperities and on top of them. The ice adhesion values of PTFE-Fill- 6 and PTFE-Fill-3 were 20 ± 2 kPa and 70 ± 5 kPa, respectively. Under the same conditions, the adhesive force of ice on the bare Al substrate was 200 kPa, and the average ARF was about six. Comparing the ice adhesion of the filled samples with that of the PTFE-coated Al substrate (120 kPa) (Figure 5) indicates

that the hierarchical structure of $TiO_2$ has helped to reduce ice detachment to an average of 40 kPa (80% reduction).

To highlight the relevance of these ARF values, they were compared to the ARFs of 274 icephobic coatings developed and reported between 2003 and 2015; 67% of the coatings showed ARFs between 1 and 5, while only 23% provided ARFs above 10 [42]. Nevertheless, these values were established at lower temperatures ($-8\,^\circ$C$/-10\,^\circ$C) than in this study, and therefore, concerned less problematic forms of ice (mixed-rime versus clear ice, in this study). As a result, it could be assumed that the solution of the filled sample would be a fairly effective performing coating.

### 3.7. Mechanical Durability of the Coating

Since the efficiency of an icephobic surface depends on its durability, the mechanical durability was determined by measuring the ice adhesion after repeated icing–deicing cycles, which is a more accurate test technique than detachment by melting the ice because it can severely damage the substrate [47]. The results of ice shear adhesion strength versus the number of cycles are shown in Figure 6, and the error bars represent the standard deviation performed for different measurements.

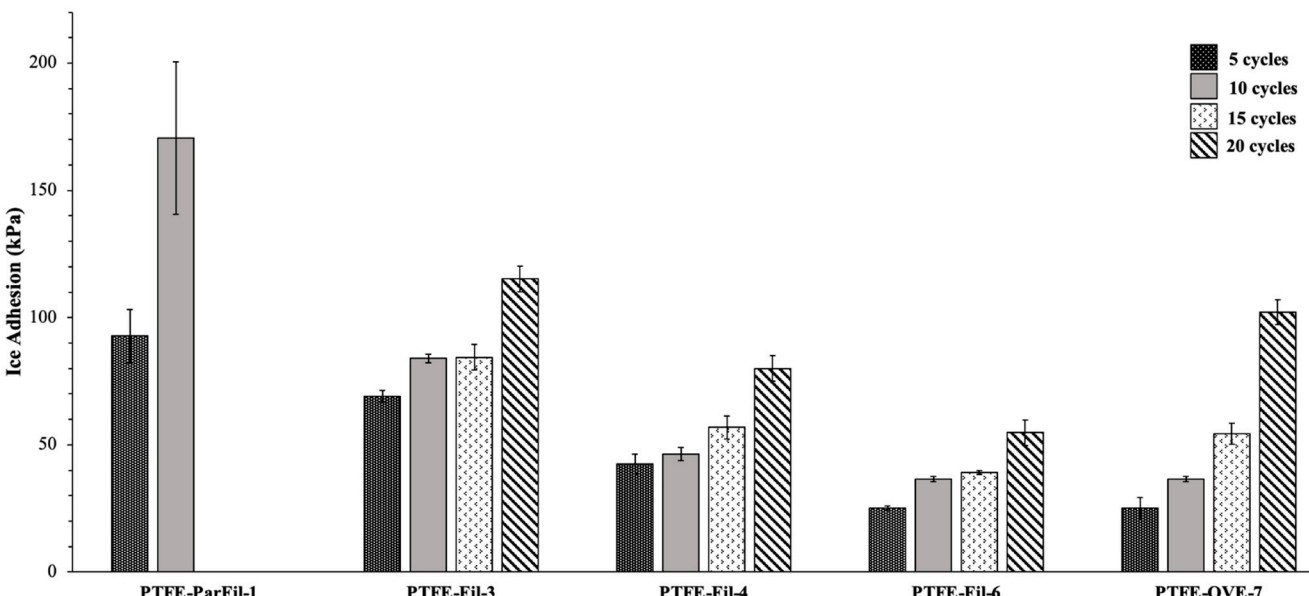

**Figure 6.** Average ice adhesion measurements for every five cycles.

The behavior of the partly filled sample was different from that of the other samples, as shown in Figure 6. During the first ten cycles, the change in ice adhesion on the partially filled samples was more significant. The adhesion strength quickly increased after five cycles, and finally, very high adhesion values (about 170 kPa) showed that the PTFE solid lubricant films failed.

It has been shown that some surface roughness parameters, including the height of features, the width of the gap between features, and the presence of dual-scale roughness, affect the wettability and icephobicity of the surface [48,49]. Therefore, the roughness parameters were also measured every five cycles and compared with the initial state of the samples to study their effect on the durability of the film. Table 6 shows the roughness parameters of PTFE-ParFill-1, every five cycles before the film is severely damaged. When compared to the initial values, Table 6 indicates an increase in most roughness parameters, such as $R_z$, $R_a$, $R_q$, and $R_{ku}$. The $R_{ku}$ values are less than three before the initial ice adhesion, but they increase to roughly three after five cycles and more than three after ten cycles. In addition, the increase in the number of valleys on the surface is confirmed by the $R_{sk}$ value reduction to more negative values during deicing cycles.

**Table 6.** Roughness parameters of sample PTFE-ParFill-1 during mechanical deicing.

| No. of Cycles | $R_z$ (µm) | $R_a$ (µm) | $R_q$ (µm) | $R_{Sk}$ | $R_{ku}$ |
|---|---|---|---|---|---|
| Initial | $18 \pm 0.2$ | $4.23 \pm 0.2$ | $5.1 \pm 0.7$ | $0.16 \pm 0.01$ | $2.43 \pm 0.1$ |
| 5 | $20.2 \pm 0.7$ | $4.90 \pm 0.1$ | $7 \pm 0.4$ | $-0.09 \pm 0.02$ | $2.98 \pm 0.1$ |
| 10 | $22.5 \pm 0.9$ | $5.84 \pm 0.2$ | $8.9 \pm 0.8$ | $-0.10 \pm 0.01$ | $3.43 \pm 0.1$ |

The partially filled samples were characterized, and after seven deicing cycles, most $TiO_2$ tips lost their PTFE solid lubricant films (shown by yellow circles in Figure 7a). More specifically, 5% of Figure 7a represents damaged PTFE solid lubricant areas, while 1.6% of the surface is $TiO_2$ texture. The ice formation and detachment occurred on the bare $TiO_2$ cauliflower structures. Then, ice separation may have caused severe damage to the nanostructure of the cauliflowers and led to the detachment of $TiO_2$ textures, as seen in Figure 7a with red circles and Figure 7b with glossy patches.

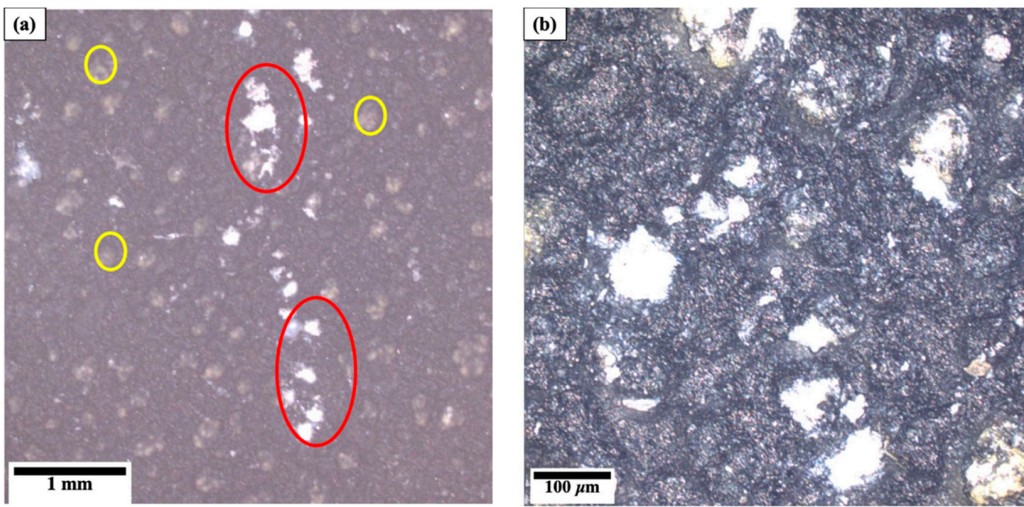

**Figure 7.** The surface of partially filled samples: (**a**) Damage to PTFE solid lubricant film (yellow circles) and $TiO_2$ texture (red circles) after seven cycles, magnification 5×; (**b**) $TiO_2$ cauliflower detachment after ten cycles, magnification 20×.

The overfilled samples showed very low ice adhesion for at least ten cycles (36 kPa). However, the shear adhesion began to rise when the icing–deicing cycles increased to more than 10 (Figure 6). The ice adhesion of PTFE-OVE-7 was an average of 54 kPa after 15 cycles, and then it exceeded 100 kPa for more cycles. According to Table 7, the roughness values of sample PTFE-OVE-7, after five cycles, are very similar to the initial ones, then increase slightly after ten cycles of ice detachment. Compared to the partially filled samples, the roughness result of PTFE-OVE-7 in Table 7 shows minor increases in $R_z$, $R_a$, and $R_q$, confirming the better coverage of the solid lubricant film on the $TiO_2$ coating after 20 cycles of ice detachment.

**Table 7.** Roughness parameters of sample PTFE-OVE-7 during mechanical deicing.

| No. of Cycles | $R_z$ (µm) | $R_a$ (µm) | $R_q$ (µm) | $R_{Sk}$ | $R_{ku}$ |
|---|---|---|---|---|---|
| Initial | $2.8 \pm 0.5$ | $1 \pm 0.1$ | $1.2 \pm 0.1$ | $-0.05 \pm 0.01$ | $2.8 \pm 0.01$ |
| 5 | $2.9 \pm 0.7$ | $1.3 \pm 0.1$ | $1.5 \pm 0.4$ | $-0.06 \pm 0.02$ | $2.8 \pm 0.03$ |
| 10 | $3.1 \pm 0.9$ | $1.6 \pm 0.3$ | $1.8 \pm 0.2$ | $-0.08 \pm 0.01$ | $2.9 \pm 0.02$ |
| 15 | $3.9 \pm 0.4$ | $2 \pm 0.3$ | $2.2 \pm 0.2$ | $-0.08 \pm 0.03$ | $3.0 \pm 0.02$ |
| 20 | $4.8 \pm 0.5$ | $4.3 \pm 0.1$ | $4.5 \pm 0.3$ | $-0.09 \pm 0.02$ | $3.2 \pm 0.03$ |

Although $R_{sk}$ values are very close to zero in all cycles, they also gradually reduce to more negative values, indicating an increase in the number of valleys on the surface. The $R_{ku}$ value also increases gradually with the number of deicing cycles. Confocal microscopy characterization of the solid lubricant surface after 20 cycles revealed the presence of yellow dots (4% of the surface) that confirmed film separation on top of the cauliflowers, as illustrated in Figure 8.

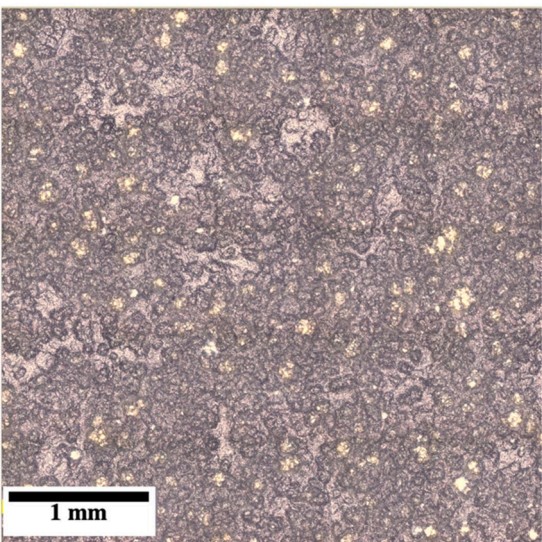

**Figure 8.** Surface of the overfilled PTFE solid lubricants sample, showing film detachment on the top of cauliflowers (yellow dots) after 20 cycles.

Finally, the filled samples demonstrated considerable mechanical durability and out-performed the others in terms of durability. The ice adhesion values remained close to 50 and 75 kPa after 10 and 15 cycles, respectively, and increased to roughly 80 kPa after 20 mechanical deicing cycles. Sample PTFE-Fil-6 was also the most durable icephobic sample, and its average ice adhesion was below 50 kPa for up to 20 cycles. Throughout the deicing cycles, the surface roughness values of sample PTFE-Fil-6 were measured (Table 8), and $R_p$ and $R_v$ did not significantly change.

**Table 8.** Roughness parameters for sample PTFE-Fil-6 during mechanical deicing.

| No. of Cycles | $R_z$ (μm) | $R_a$ (μm) | $R_q$ (μm) | $R_{Sk}$ | $R_{ku}$ |
|---|---|---|---|---|---|
| Initial | $4.2 \pm 0.4$ | $1.2 \pm 0.1$ | $1.4 \pm 0.02$ | $0.05 \pm 0.01$ | $2.96 \pm 0.01$ |
| 5 | $4.2 \pm 0.6$ | $1.1 \pm 0.2$ | $1.3 \pm 0.1$ | $0.09 \pm 0.02$ | $2.7 \pm 0.05$ |
| 10 | $4.1 \pm 0.9$ | $1.2 \pm 0.3$ | $1.4 \pm 0.2$ | $0.06 \pm 0.01$ | $2.4 \pm 0.07$ |
| 15 | $4 \pm 0.5$ | $1.4 \pm 0.3$ | $1.5 \pm 0.2$ | $0.03 \pm 0.01$ | $3.1 \pm 0.09$ |
| 20 | $4.1 \pm 0.3$ | $1.4 \pm 0.3$ | $1.5 \pm 0.7$ | $0.04 \pm 0.02$ | $2.9 \pm 0.04$ |

To investigate the damage and delamination of the PTFE solid lubricant film, the surface of sample PTFE-Fil-6 was examined using the confocal microscope at two different magnifications, as shown in Figure 9. No damage to the surface and cauliflowers can be seen at the 5X magnification in Figure 9a, but the higher magnification shows minor removal of PTFE solid lubricants from the top of some cauliflowers after 20 cycles of deicing (Figure 9b). The PTFE solid lubricant film detachment mostly occurred in the front part of the sample, where higher stress was concentrated due to the applied force during ice detachment. However, these minor film detachments did not significantly affect the ice adhesion values.

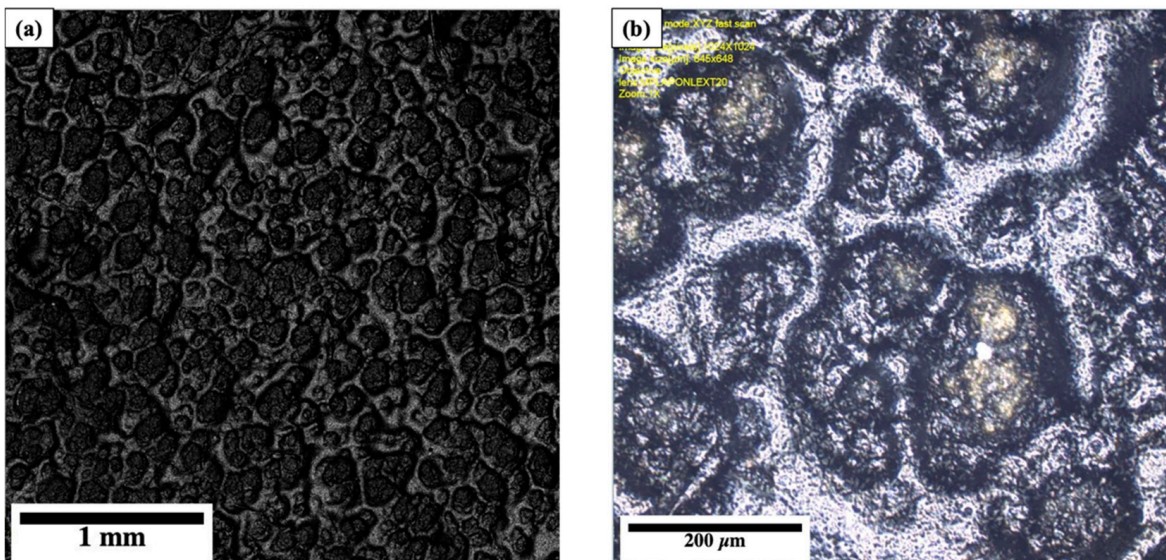

**Figure 9.** The surface of sample PTFE-Fil-6 after 20 cycles of mechanical deicing: (**a**) At 5× magnification; (**b**) at 20× magnification, showing minor detachment at the top of TiO$_2$ cauliflowers.

## 4. Discussion

### 4.1. Effect of Wetting Parameters on Ice Adhesion

Table 3 shows that the filled samples have higher hydrophobicity and better water repellency (WCA 131° ± 7 and CAH = 22° ± 0.4) than the sprayed TiO$_2$ coating. This result was in agreement with other studies that used thin hydrophobic layers (e.g., stearic acid and DLC/SiO$_x$) to functionalize SPS TiO$_2$ coating and provided hydrophobic top layers, with static WCA values of 97° ± 1 and 159° ± 1, respectively [18,20]. However, it appears that the advantageous TiO$_2$ characteristics do not really affect the wettability results of the overfilled samples since their wetting properties (Table 3) are similar to a pure, flat PTFE solid lubricant surface (WCA = 94° ± 2 and CAH = 48° ± 6) [27]. This could reasonably be explained by the fact that the surface structure of the overfilled sample is a relatable PTFE solid lubricant layer. Furthermore, the results of Figure 5 show that the hydrophobic PTFE film reduced the solid–liquid interaction in all samples and enhanced the effect of the hierarchical structure of TiO$_2$ coating. The combination of dual roughness and low wettability also weakened solid/ice bonds at the interface and helped to develop an icephobic surface with a lower ice adhesion, which was in agreement with similar studies that tried to reduce the surface energy of textured coatings [50].

### 4.2. Effect of Surface Roughness on the Ice Adhesion

Several roughness parameters were actively investigated (Table 4) for two reasons. First, since the PTFE-coated samples have the same surface chemistry, their various wettability (Table 3) and anti-icing behaviors (Figure 5) can be related to differences in surface roughness characteristics. Second, by comparing the R$_v$, R$_p$, and R$_z$ values of the bare TiO$_2$ and PTFE-coated samples, it is possible to assess the film distribution on top of the cauliflowers and the spaces between the cauliflowers, which can have a considerable effect on ice adhesion. In other words, a decrease in R$_v$ values can primarily result in a decrease in R$_z$ values. When the R$_v$ value (the valley to the mean line) is lower, the mean line also moves and causes a reduction in the R$_p$ value (the peak to the mean line). The negligible change in R$_p$ values might be due to the relatively tiny area of cauliflower tips compared to the valley between them and their ability to be protected by a thin film. Therefore, any reduction in R$_v$ values might be the main factor causing the decline in R$_z$ and indicate that the PTFE solid lubricant film can fill the valleys between TiO$_2$ coating asperities. Figure 2 also supports a different distribution of the PTFE solid lubricant film on the samples under various vibration and heating conditions.

Due to the high WCA and low CAH values of the partially filled samples (Table 3) and their high roughness parameters (e.g., $R_a$ and $R_q$) (Table 4), it seems that water droplets can develop stronger mechanical interlocking on the surface of the partially filled samples than the filled or overfilled samples. Therefore, higher roughness parameters would seem to indicate a worst-case scenario for ice adhesion when comparing icephobic properties, which is in line with prior studies [32]. However, the coverage of the PTFE solid lubricant helped to reduce the adhesion compared to that on the bare $TiO_2$ sample. Although PTFE solid lubricants were applied unevenly and without complete coverage of cauliflowers in the partially filled samples, the film covered the area between the asperities and provided a smooth contact area with ice, as illustrated in Figure 2a.

The filled samples produced without heating (notably sample PTFE-Fil-6) showed a less significant difference between $R_v$ and $R_p$, values, resulting in smaller $R_z$ values among the filled samples, as shown in Table 4. Therefore, uniform distribution of the PTFE solid lubricant film on the $TiO_2$ coating and between the asperities was achieved (Figure 2b). Compared to the partially filled samples, the $R_a$, $R_z$, and $R_q$ values (Table 4) of the filled samples were much lower. As a result, the filled samples could be ideal for ice adhesion due to their low wettability (Table 3) and roughness characteristics when the PTFE solid lubricant film uniformly covers the asperities and top layers of nano features. This appears to be in agreement with the results of Figure 5.

The PTFE solid lubricant film generated a bumpy but smooth surface in the overfilled samples when it completely covered the texture of the $TiO_2$ coating. Lower ice/solid interfacial strength and easier detachment can be due to the stress-rising effect of the bumpy interface with low roughness parameters, such as $R_a$ and $R_q$ (Table 4), which decreases ice adhesion. This is consistent with the results of other studies [51–53]. In parallel, the works of Farahani et al. [27] also provided a mechanism that could explain ice detachment from PTFE lubricant films, but in the circumstances of this study, this mechanism would apply irrelevantly of the underlying $TiO_2$ structure, and is not explored in further detail here.

Moreover, it has been shown that $TiO_2$ cauliflower features, covered by low surface energy PTFE solid lubricant film, can help to entrap air pockets between the water droplets and the substrate during freezing [8], which can provide a Cassie–Baxter wetting mode [6]. As previously discussed, these air pockets at the interface can act as defects and stress concentrators, facilitating interfacial crack propagation and eventually making ice detachment easier [54]. However, a large negative value of $R_{sk}$ can also lead to more contact area between the ice and the substrate, increasing ice adhesion when very small-sized water droplets impact the surface at high speed [55]. Accordingly, the bare $TiO_2$ sample with high surface energy (Table 3) and high roughness (Table 4) led to considerable ice adhesion (Figure 5). Moreover, a combination of the filled samples with the dual-scale roughness of $TiO_2$ coating (shown in Figure 3) might be helpful to trap the air pockets and can hold water droplets on the top. It has also been reported that the air trapped within the hierarchical roughness acts as a thermal insulation layer, reducing the solid/liquid contact area. Air pockets might decrease heat transfer and act as barriers to heterogeneous ice nucleation [45,56]. The delay in surface temperature reduction was also confirmed by the results of Figure 4. As the droplets freeze on the peaks, water vapor within the valleys reduces. The existence of dry or under-saturated valleys at the interface can encourage evaporation of water collected within the valleys and delay further condensation, thereby reducing ice formation within the valleys [57,58]. As a result, the valleys transform to voids and defects at the interface, which can slow down the formation of ice crystals, act as stress increasers at the interface, and prevent the formation of strong ice interlocking. This is consistent with the results shown in Figure 5.

In addition, the height and shape of the asperities can be estimated by another surface parameter called kurtosis ($R_{ku}$) [8,59,60]. It has been shown that the actual surface area at the ice–substrate interface is highly dependent on the kurtosis, and the higher the $R_{ku}$, the higher the interface contact area [32]. A larger contact area between the substrate and the

freezing water droplets can lead to stronger bonding and ice adhesion [43]. The surface of bare $TiO_2$ samples showed considerable divergence from the normal distribution, and their kurtosis values were larger than three, which showed the highest ice adhesion (Figure 5). However, all filled and overfilled samples had kurtosis values close to three (Table 4), lower contact area after freezing, and less ice adhesion (Figure 5), yet showed a variety of peak-to-valley heights, showing that other parameters may still need to be studied to accurately assess how the various roughness parameters effectively impact ice adhesion to the surface.

The results of the ice accretion test showed that a combination of PTFE solid lubricant and dual-scale roughness of $TiO_2$ coating could stimulate detachment of the impinging droplets, which dynamically decreases the chance of heterogeneous ice nucleation and reduces the total amount of the developed ice, as observed in other studies [61]. Nevertheless, while the temperature and impact angles had influences that could be characterized, another crucial factor was the change in the airstream speed, which had two conflicting effects on the overall mass of the accumulated ice. The first effect was that more water droplets were able to hit the surface, which resulted in more ice formation at higher airstream speeds, while all other factors remained the same. The other effect was that higher airstream speed helped separate the water droplets from the surface and increased the quantity of smaller droplets that could diverge from the sample, and both effects led to reductions in the total weight of the formed ice layer. Since the effect of the airstream is difficult to comprehend, it should be investigated in combination with other parameters.

*4.3. Mechanical Durability*

The mechanical durability results, shown in Figure 6, seem to show a close correlation with the surface roughness parameters and physical condition of the PTFE solid lubricant film after each deicing cycle. A possible explanation for the differences in ice adhesion during several deicing cycles of the samples could be the anchoring action of the $TiO_2$ substrate that holds the solid lubricant film. In other words, since $TiO_2$ features showed a significant anchoring effect on the filled samples and relatively weaker van der Waals forces held the solid lubricant layers together, the adhesion between PTFE solid lubricants and $TiO_2$ features could be stronger than the adhesion between the film and the ice. On the other hand, the required anchoring between the film and the substrate was not provided for the other samples due to different vibration and heating times during the film deposition.

According to Table 6, the surface of the partially filled sample has considerably changing peak-to-valley distribution, as highlighted by the changing values of $R_{ku}$ after five, and then ten deicing cycles (from 2.43 to 3.43). Lower $R_{sk}$ values and higher $R_z$ values after cycling likely indicate that the PTFE solid lubricant film was detached from the cavities during the dicing cycles. Therefore, ice adhesion would suddenly increase due to an increase in the total contact area at the interface. Deeper ice formation within the asperities also occurs, leading to stronger ice interlocking [62]. The damage to the cauliflower-like structure and PTFE solid lubricant film, shown in Figure 7, also supports the improved ice adhesion results of Figure 6.

Although the overfilled samples showed considerably low ice adhesion, they did not indicate acceptable mechanical durability, as shown in Figure 6. For these samples, the film was deposited with no vibration or heating, which is the only factor that sets it apart from other samples (Table 1). The separation of the PTFE solid lubricant film from the tips of the cauliflower structure after 20 cycles can be confirmed by Figure 8. Table 7 also confirms that the film is worn off when, after roughly 20 cycles, $R_z$ increases from 2.8 $\mu$m before deicing to 4.8, $R_{sk}$ is reduced, and $R_{ku}$ is larger. A larger $R_{ku}$ and more negative values of $R_{sk}$ can increase the likelihood of impinging and freezing water droplets within the deep areas. Consequently, the ice has a larger surface area when it comes into contact with the PTFE solid substrate, in contrast to primary deicing cycles. Therefore, the adhesion between the PTFE film and the substrate is not as strong as the interfacial strength of ice/solid lubricant, leading to a rapid rise in ice adhesion as some of the lubricant films are worn away.

Finally, the filled samples showed the highest mechanical durability, as shown in Figure 6, due to their highest anchoring to the $TiO_2$ coating. According to the roughness parameters listed in Table 8, such as $R_z$, enough film remains on the $TiO_2$ cauliflower structure, and there is no loss of PTFE solid lubricant during deicing. Furthermore, the stability and durability of the film can be shown by the consistency of $R_a$ and $R_q$ after 20 cycles, shown in Table 8. Nevertheless, the detachment of some PTFE solid lubricant in the front of the sample (Figure 9) could indicate a slow beginning to the failure of the PTFE lubricant, and it would be interesting to perform the mechanical durability tests for a higher number of cycles. Furthermore, it was observed that the amount of damaged surface structure somewhat correlated with decreasing mechanical durability, as could be observed when comparing the partially filled sample surface after 7 cycles (6.6% damaged surface) (Figure 7a), the overfilled sample surface after 20 cycles (4%) (Figure 8), the filled sample surface after 20 cycles (<1%) (Figure 9) and the mechanical durability results (125 kPa, 100 kPa, and 55 kPa, respectively) (Figure 6). Therefore, it may be suggested that more consistent characterization of sample surfaces after each cycle could help lead to a correlation between surface condition (and potentially surface roughness) and the potential mechanical durability of the coated samples, which could also be the focus of future work.

## 5. Conclusions

In this study, PTFE solid lubricant film was applied on top of suspension plasma-sprayed $TiO_2$ coating with the objective of improving the water repellency of SPS $TiO_2$ coatings, and therefore, the potential ice-phobic properties of these coatings. The PTFE was applied onto the $TiO_2$ coatings by brushing different amounts of PTFE, followed by different vibration and heating times, that significantly affected the distribution of the PTFE solid lubricant film. The surfaces were characterized, and the "filled" samples showed the highest wettability and water mobility with a uniform distribution on top and between the dual-scale roughness features of $TiO_2$. Furthermore, the properties of the PTFE solid lubricant films were tested via three tests, which focused on the amount of accumulated ice, ice shear adhesion and mechanical durability of the samples. The amount of accumulated ice for the "filled" sample was significantly lower than for the other samples.

By comparing the results, the film thickness and prior information on SPS $TiO_2$ coatings (dual-scale structure), it was possible to establish that, on the one hand, a thicker film ("overfilled" sample) may produce more protection to the substrate, but that the film would reduce the beneficial effects of the nanoscale $TiO_2$ surface characteristics. On the other hand, a thinner film ("partially filled" sample) does not provide any ice adhesion reduction, as the ice rapidly comes into contact with the nanoscale $TiO_2$ structure. In addition, both of these PTFE film thicknesses also lead to relatively low mechanical durability.

However, the filled samples with the highest WCA and the lowest CAH values showed considerable durability with an acceptable ice adhesion for up to 20 cycles. The low ice adhesion of the filled samples can be attributed to the $TiO_2$ dual-scale roughness that trapped air, acted as a stress riser at the interface after freezing, and led to very low mechanical interlocking and ice adhesion. The highest durability of filled samples might result from the significant anchoring effect between the PTFE solid lubricant film and $TiO_2$ features due to the vibration applied during film deposition.

This filled sample PTFE-solid lubricant structure on SPS $TiO_2$ coating, therefore, appears to be very promising in terms of creating a valuable anti-icing solution. Nevertheless, the current study focused on different coating types, and only on a certain number of icing–deicing cycles for mechanical durability (20), with some defects appearing near the end of the durability study, indicating that further mechanical property studies could be required.

**Author Contributions:** Methodology, data curation and writing—original draft preparation, E.F.; writing—review and editing, A.C.L. and A.M.; Conceptualization, supervision, resources and funding acquisition, P.S., C.M. and A.D. All authors have read and agreed to the published version of the manuscript.

**Funding:** This research received no external funding.

**Institutional Review Board Statement:** Not applicable.

**Informed Consent Statement:** Not applicable.

**Data Availability Statement:** All data that support the findings of this study are included within the article.

**Conflicts of Interest:** The authors declare no conflict of interest.

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
