# Peer review of "Ice Adhesion Evaluation of PTFE Solid Lubricant Film Applied on TiO2 Coatings"

_coatings, doi:10.3390/coatings13061049_

Round 1

Reviewer 1 Report

The author proposes a simple method for the preparation of a duplex coating based on titanium dioxide/polytetrafluoroethylene (PTFE) solid lubricant filmAnd it is evaluated for ice-sparing and mechanical durability on the corresponding substrates. This research is meaningful and interestingHowever, there are still some problems with the paper in terms of performance characterisation and mechanism interpretation. In general, the overall quality of the manuscript needs further improvement. A detailed classification of the problems is given below, which may also help the authors to write a better manuscript.

1. There is a distinction between hydrophobic and anti-icing mechanisms, and the authors need to consider the differences and connections between the two carefully. In addition, the authors should provide a more complete and specific explanation of the anti-icing mechanism in order to improve the academic standard of the article.

2. In section 3.7, only SEM was used to observe the structural damage of the coating, and it is not sufficient. A more in-depth research including material failure and mechanical durability is desired. The surface composition analysis, and element distribution in this area should be characterized. 

3. Many similar studies using TiO2 or hydrophobic materials as anti-icing coatings, and the innovation of your research should be further highlighted. 

4. This work provides a double-coating method to to increase roughness and reduce surface energyHowever, the adhesion between multi-layer coatings is often poor. What are the specific interfacial groups contributing to the enhanced interfacial bonding in this work?

5. Compared with others, what are the advantages of this article in terms of anti-icing efficiency and recyclability?It is suggested that adding comparative data analysis would be more obvious.

 There are some formatting errors in the references that need to be checked and corrected.

Reviewer 2 Report

This paper deals with development of a new method to reduce ice adhesion. Many useful results were obtained from various measurements.

This study will be worthy to be published on this journal after some corrections.

Some suggestions are as follows.

Comment 1: Line 202 “IWT”

You use many abbreviations in the present article. It will be convenient for you to use various abbreviations in your experiment notebook. However, other researchers use not abbreviations but ”water contact angle”, “contact angle hysteresis” and so on in many articles.

Because you explain an abbreviation properly, I respect the intention of the author that many abbreviations are used in the present article.

Is the word “IWT” mean ”icing wind tunnel”? Where is the explanation of “IWT”?

Comment 2: Abstract, 2. Experimental Method, Figure 6, 3.6. Shear adhesion strength of ice

You compare the results of various treated samples with the result of bare TiO2 and polished Al in Figure 6. You also write ”compared to that on the bare aluminum substrate” in “Abstract”.

The substrate used in the present study is “stainless steel” as written in ” 2. Experimental Method, 2.1. Sample preparation and PTFE coating application” (Line 122).

The change of the results of stainless steel before and after the treatment should be compared. That is, bare stainless steel should be used as a standard material.

Comment 3: Conclusion

Concrete values and detailed results should not be written in "Conclusion". The results obtained in the present study should be written clearly in "Conclusion".

Please explain what kind of material was provided by what kind of method in this study.

Comment 4: Other small comments

Line 269, Is “Fig. 5c” right? “Fig. 3c”?

Line 399, “~ The adhesion ~” should be “~ the adhesion ~”.

Reviewer 3 Report

This manuscript describes the impact of differing amounts of polytetrafluorethylene (PTFE) coating onto titanium dioxide on the formation and adhesion of ice on the surface. The methodology is reasonable and a set of useful experiments conducted from which some useful information has been obtained on the impacts of different coating amounts and methods on resistance to ice formation and on mechanical stability through icing and deicing cycles. However, there are a number of parts of the manuscript where more details and clarification are needed. Specific points for the authors to address are detailed below (assume all of them require changes to the manuscript).

1.       Throughout the manuscript you mention the term “dual-scale” but it is not apparent what you mean here. Whilst the titanium dioxide certainly has surface roughness, it is not clear that it has two distinct morphologies together within the same surface. Some clarification and reconsideration of this terminology is needed.

2.       Page 3 line 115 Define “IWT” at first use in the manuscript.

3.       Table 1. It would be clearer to replace the dashed lines with “0” as the vibration and/or heating times were 0 minutes.

4.       Table 2. These details can be provided in the main text, so this table is superfluous.

5.       Figure 2. These images come from reference 19 and as such should not be reprinted in this manuscript but rather an appropriate mention and citation of the reference should be given in the main text.

6.       Table 3. Is the WCA value for titanium dioxide taken from a literature reference or experimentally determined by the authors? If it is from the literature, then a supporting reference should be provided. If it is experimentally determined then errors should also be included.

7.       For a number of the experiments the authors have not clarified the number of replicates used. Details of the number of replicates used should be provided alongside all results summarised in the section 3 text and/or associated tables and figures.

8.       The parameter Rz does not appear to have been defined properly in the manuscript. Please ensure this is defined, ideally in section 2.2 alongside the other R parameters.

9.       Page 9 lines 320-324 “The Ra values . . . were both preserved.” This is an example of where it is important for the reader to understand whether you are looking at the same points on the same sample before and after PTFE deposition or are taking averages across a large number of different points in the sample before and after deposition. Details and clarification are needed here.

10.   Figure 5. How consistent are these results across multiple samples?

11.   Figure 5. Which PTFE coating protocol was used?

12.   Page 10 lines 340-341 “. . . with a 280-second delay” It is not clear how this number relates to Figure 5. 280 s is roughly the time taken for the bare titanium dioxide to reach approximately -1 °C but not the PTFE-coated surface. This description needs to be reconsidered.

13.   Page 11 lines 369-370 “. . . considering the fact . . . ice to detach . . .” This statement needs a supporting reference.

14.   Figure 6. How was the PTFE coated onto the aluminium in Al + PTFE film? Succinct details should be provided in the manuscript.

15.   Page 14 lines 481-483 “The PTFE solid . . . ice detachment.” This is an interesting aspect. How much variability have you seen across different parts of the samples with the other coatings? Is this spatial variability significant or negligible when considering the overall properties of the samples? Some discussion of this in the manuscript would be useful.

The quality of English in the manuscript is mostly adequate. There are just some minor grammatical errors that need to be corrected during the editing process.

Round 2

Reviewer 1 Report

The authors have made reasonable changes to resolve each of the previous questions, and my concerns have been addressed. I think this paper is suitable for publication in Coatings and recommend it for acceptance.

The author's language is relatively concise and the experimental findings are clearly described.

Author Response

Thank you for your review and feedback to help improve the quality of this manuscript.

Reviewer 2 Report

This article was revised enough, so it can be published.

Author Response

Thank you for your time and feedback to help improve this manuscript. 

Reviewer 3 Report

The authors have made a number of appropriate changes to the manuscript in response to my earlier review and I believe this has improved the manuscript. There are however two areas where I remain in disagreement with the authors. These are:

1. I feel Table 2 is superfluous and, given the large number of tables and limitations on journal space, the data could easily be quoted in the main text.

2. It is my opinion that for original research articles (i.e. not reviews) reproduction of figures from a previous publication is seldom either appropriate or merited. I therefore still feel that Figure 2 should be removed as it is essentially padding out your current paper with material from a previous paper.

The English language quality is adequate. A minor editorial check is likely to be sufficient.

Author Response

The authors would like to thank the reviewer for their feedback and open discussion on some elements from this manuscript. For the second round of reviews, the different points have been addressed as follows : 

Point 1. I feel Table 2 is superfluous and, given the large number of tables and limitations on journal space, the data could easily be quoted in the main text.

The authors agree that the test conditions may be added in the main text, but they also believe that the test conditions are better seen in the table. As a result a section has been added to the text to take this into account : "The tests were conducted during 180s at a temperature of -3 °C, an airspeed of 25 m/s, with an MVD of 30 µm and LWC of 0.8 g/m3 : this contributed to generating 2 to 3 mm of ice on the surface of the samples."

Point 2. It is my opinion that for original research articles (i.e. not reviews) reproduction of figures from a previous publication is seldom either appropriate or merited. I therefore still feel that Figure 2 should be removed as it is essentially padding out your current paper with material from a previous paper.

Figure 2 has been removed and all other figure numbers corrected accordingly.